# Hepatic DKK1-driven steatosis is CD36 dependent

Zhen Yang[1,2] , Xinping Huang[1,2] , Jiaye Zhang[1], Kai You[1], Yue Xiong[1], Ji Fang[1], Anteneh Getachew[1], Ziqi Cheng[1,2], Xiaorui Yu[1,3], Yan Wang[1], Feima Wu[1], Ning Wang[1], Shufen Feng[7], Xianhua Lin[1], Fan Yang[10], Yan Chen[1], Hongcheng Wei[7], Yin-xiong Li[1,2,4,5,6,8,9]

**Nonalcoholic fatty liver disease (NAFLD) is prevalent worldwide; about 25% of NAFLD silently progress into steatohepatitis, in which some of them may develop into fibrosis, cirrhosis and liver failure. However, few drugs are available for NAFLD, partly because of an incomplete understanding of its pathogenic mechanisms. Here, using in vivo and in vitro gain- and loss-of-function approaches, we identified up-regulated DKK1 plays a pivotal role in high-fat diet–induced NAFLD and its progression. Mechanistic analysis reveals that DKK1 enhances the capacity of hepatocytes to uptake fatty acids through the ERK-PPARγ-CD36 axis. Moreover, DKK1 increased insulin resistance by activating the JNK signaling, which in turn exacerbates disorders of hepatic lipid metabolism. Our finding suggests that DKK1 may be a potential therapeutic and diagnosis candidate for NAFLD and metabolic disorder progression.**

## Introduction

Nonalcoholic fatty liver disease (NAFLD), also known as metabolic-associated fatty liver disease (Eslam et al, 2020), is a syndrome characterized by excessive lipid deposition in hepatocytes because of factors other than alcohol (Powell et al, 2021). It is a manifestation of metabolic syndrome in the liver and is usually accompanied by obesity (Cusi, 2012), insulin resistance (Yu et al, 2013) and diabetes mellitus. Moreover, NAFLD may progress to nonalcoholic steatohepatitis (NASH), liver fibrosis, scarring (cirrhosis) and even liver cancer (Hardy et al, 2016). NAFLD has become the most common chronic liver disease globally (Powell et al, 2021), although limited effective clinical pharmacological therapies have been approved by the FDA yet (Friedman et al, 2018). Therefore, it has become a priority to investigate the pathological mechanism of NAFLD in depth and find potential therapeutic targets.

Excessive intracellular lipid deposition (mainly triglycerides [TG]) in hepatocytes is a major feature of NAFLD and also permeates the entire NAFLD disease spectrum (Pelusi & Valenti, 2019). Hepatic lipid metabolism comprises several important processes, including fatty acid uptake, β-oxidation, secretion and de novo lipogenesis (Tanoli et al, 2004; Postic & Girard, 2008; Lambert et al, 2014), which all influence the balance of hepatic lipid homeostasis (Neuschwander-Tetri, 2010). In general, hepatic steatosis is an outcome of the uptake of fatty acids and de novo lipogenesis surpassing fatty acid oxidation and export.

A scavenger receptor CD36 (FAT/CD36) located on hepatocytes plays multiple physiological roles including receptor-mediated free fatty acid uptake. Pathologically, CD36 participates in lipid accumulation, oxidative stress and inflammation. Increased fatty acid uptake has been directly related to the increased hepatic CD36 under high-fat diet (HFD)–induced hepatic steatosis in mice (Silverstein & Febbraio, 2009; Miquilena-Colina et al, 2011). The same observation was also confirmed in a mice model, where adenovirus-delivered overexpression (OE) of CD36 in the liver significantly increased hepatic fatty acid uptake and fat accumulation even under chow diet condition (Koonen et al, 2007). Nonsense mutations in CD36 are also being associated with insulin resistance, familial type 2 diabetes, CD36-mediated ER stress and inflammatory signaling (JNK, NF-κB) implicated in diet-induced obesity and reduced insulin sensitivity (Furuhashi et al, 2003; Lepretre et al, 2004; Karunakaran et al, 2021).

In addition, insulin resistance has been shown to trigger lipid accumulation in the early stages of NAFLD (Guo, 2014). Under insulin-resistant condition, the inhibitory effect of insulin on adipose tissue catabolism is weakened (Perry et al, 2014), leading to elevated serum FFA, TG, and TC levels and gluconeogenesis and TG accumulation in the liver. Insulin resistance in the liver is mainly involved with the defective signaling transduction of PI3K-AKT and Ras-MAPK pathway (Tilg & Moschen, 2008) resulted in insulin and glucose tolerance, further leading to the accumulation of TG and aggravating the development of NAFLD.

[1]Center for Health Research, Guangzhou Institutes of Biomedicine and Health, Chinese Academy of Sciences, Guangzhou, China   [2]University of Chinese Academy of Sciences, Beijing, China   [3]School of Life Sciences, University of Science and Technology of China, Hefei, China   [4]Key Laboratory of Stem Cell and Regenerative Medicine, Guangzhou Institutes of Biomedicine and Health, Chinese Academy of Sciences, Guangzhou, China   [5]CAS Key Laboratory of Regenerative Biology, Guangzhou Institutes of Biomedicine and Health, Chinese Academy of Sciences, Guangzhou, China   [6]Guangdong Provincial Key Laboratory of Biocomputing, Guangzhou Institutes of Biomedicine and Health, Chinese Academy of Sciences, Guangzhou, China   [7]Department of Gastroenterology, First Affiliated Hospital of Jinan University, Guangzhou, China   [8]State Key Laboratory of Respiratory Disease, Guangzhou, China   [9]China-New Zealand Joint Laboratory on Biomedicine and Health, Guangzhou, China   [10]Ministry of Education CNS Regeneration Collaborative Joint Laboratory, Guangdong-Hongkong-Macau Institute of CNS Regeneration, Jinan University, Guangzhou, China

Correspondence: li_yinxiong@gibh.ac.cn

Dickkopf-1 (DKK) is a secreted potent inhibitor of Wnt signaling; it plays an essential role in Spemann organizer; craniofacial structures; kidney, limb, and hair follicle development during embryogenesis (Glinka et al, 1998; Lieven et al, 2010) and bone metabolism (Dimitri et al, 2011; Al Saedi et al, 2019). DKK1 is widely expressed in a variety of tissues and organs and involved in various biological functions, such as adipogenesis (Christodoulides et al, 2006; Gustafson & Smith, 2012; Vanella et al, 2013), cellular glycolipid metabolism (Ling et al, 2013; Xu et al, 2016) and inflammatory response (Guo et al, 2015; Chae & Bothwell, 2019). In particular, DKK1 is positively associated with central obesity (Gustafson & Smith, 2012), type 2 diabetes (Santilli et al, 2016) and atherosclerosis (Ueland et al, 2009; Li et al, 2016). Inhibition of DKK1 expression slows down HFD-induced obesity and improves insulin resistance (Gao et al, 2017). Moreover, it has been reported that r-hDKK1 enhances adipocyte differentiation efficiency by up-regulation of PPARγ and C/EBP-α and down-regulation of Wnt3a, Wnt10b, and β-catenin (Lu et al, 2016). Clinical samples revealed that serum DKK1 levels were associated with NASH (Polyzos et al, 2016), but the exact mechanism has not been fully elucidated. To address this issue, we assessed the role of DKK1 in the development of NAFLD in vivo and in vitro. Using HFD-induced NAFLD mice model, we conducted knock-down or OE of DKK1 specifically in the liver and found that DKK1 was a key player of steatosis and insulin resistance, and DKK1-driven steatosis is CD36-dependent through ERK-PPARγ signaling.

# Results

### The up-regulated hepatic expression of DKK1 significantly paralleled with steatosis

To investigate the status of DKK1 in NAFLD, liver biopsy samples from NAFLD patients (n = 5) and normal controls (n = 3) were conducted for immunohistochemical analyses, and the expression of DKK1 was significantly increased in NAFLD (Figs 1A and S1A). Then, the HFD-induced hepatic steatosis mice model was established, and a series of experiments (Fig S1B–E) were conducted to confirm the successful establishement of NAFLD. The DKK1 immunohistochemical staining observation was duplicated in the HFD-induced NAFLD mice (Fig 1B). Both mRNA (Fig 1C) and protein (Fig 1D and E) expression of DKK1 in the livers of HFD mice significantly increased (the protein, increased 63.8%, P < 0.01), and the serum DKK1 levels increased 34.8% in HFD mice compared with the counterparts of chow mice (Fig 1F, P < 0.01).

To identify the resource of produced DKK1, different liver primary cell types were isolated from HFD mice; Western blot analysis clearly revealed that even though DKK1 was expressed in some low degree in HSC and LSEC, the majority resource of DKK1 was from the hepatocytes (Fig 1G).

To confirm these observations, a mouse hepatocyte line, AML12 was used to establish steatosis by being exposed to free fatty acids (FFA, palmitic acid: oleic acid, 1:2); Western blot revealed that DKK1 was increased in a dose-dependent manner upon FFA induction (Fig 1H). Therefore, the causal relationship between DKK1 and steatosis is worth further investigation.

### The increased DKK1 was responsible for liver damage, steatosis and hyper-glyceridemia in HFD-induced NALFD

To investigate the effect of DKK1 in the development of NAFLD, we constructed liver-specific DKK1 OE and knock-down mice by injection of the adeno-associated viruses, then the NAFLD mice were induced by 20-wk HFD feed (Fig 2A). A significant body weight increase was observed in the HFD compared with the chow group, although body weight changes were not significant among HFD groups (Fig 2B). There was a marked increase or decrease in DKK1 protein expression in the liver according to the OE of DKK1 (AAV-OE-DKK1, increased 64.1%) or inhibition (AAV-sh-DKK1, decreased 51.6%), respectively, in comparison to the control group (AAV-GFP-NC) (Fig 2C).

We carefully designed this experiment with a whole set of control groups; there were eight groups in total to be conducted for the DKK1 expression manipulations in which there were parallel four groups for chow or HFD condition, including control (no virus infection, first control), AAV-GFP-NC (secondary control), AAV-OE-DKK1 (over-expressing DKK1), and AAV-sh-DKK1 (knock-down DKK1), respectively.

In chow condition, there was no steatosis or no significant difference among those four groups, whereas in HFD, all four groups were inducing steatosis with ballooning hepatocytes at different degrees. Both HFD (first control) and HFD-AAV-GFP-NC (secondary control) revealed certain level of steatosis without significant difference; however, overexpressed DKK1 promoted the steatosis; furthermore, knock-down DKK1 significantly alleviated the steatosis under HFD condition (Fig S3A).

The AAV infection did not cause steatosis in chow condition, and the steatosis had no significant difference with or without AAV-GFP infection under HFD condition. Based on these two facts, we decide to pick up four panels of represented images including the chow-AAV-GFP-NC, HFD-AAV-GFP-NC, HFD-AAV-OE-DKK1, and HFD-sh-DKK1 group mice to conduct the H&E and oil red O (ORO) analyses. All HFD-fed mice developed extensive steatosis in which the AAV-OE-DKK1 group mice had the most ballooning degeneration and lipid droplets. However, hepatic steatosis and lipid accumulation in the AAV-sh-DKK1 group were significantly alleviated (Fig 2D). The ORO results were double confirmed by biochemical analyses of cholesterol and TG content in the liver tissue (Fig 2E and F) and in serum (Fig 2G and H).

Furthermore, those observations were confirmed by qPCR examination of steatosis-related genes in which the expressions of CD36 and PPARγ were significantly changed accordingly with the DKK1 manipulations under HFD (Fig 2I).

### Gain- and loss-of-expression manipulations on cell lines confirmed the effects of DKK1 on hepatocyte steatosis

To directly investigate the effect of DKK1 on lipid metabolism in hepatocyte cells, we constructed DKK1 knockout HepG2 and AML12 cell lines using the CRISPR/Cas9 system, and those obtained knockout lines were validated by gene sequencing (Fig S2A and B) and Western blot (Fig S2C and D). In addition, DKK1 OE HepG2 and AML12 cell lines were also constructed using the lentivirus system. To screen for efficient mouse DKK1 interference target sequences, we designed a few mouse shDKK1 sequences and validated them on AML12 cells using the lentivirus system. As a verification, Western blot analyses (Fig S2J and K)

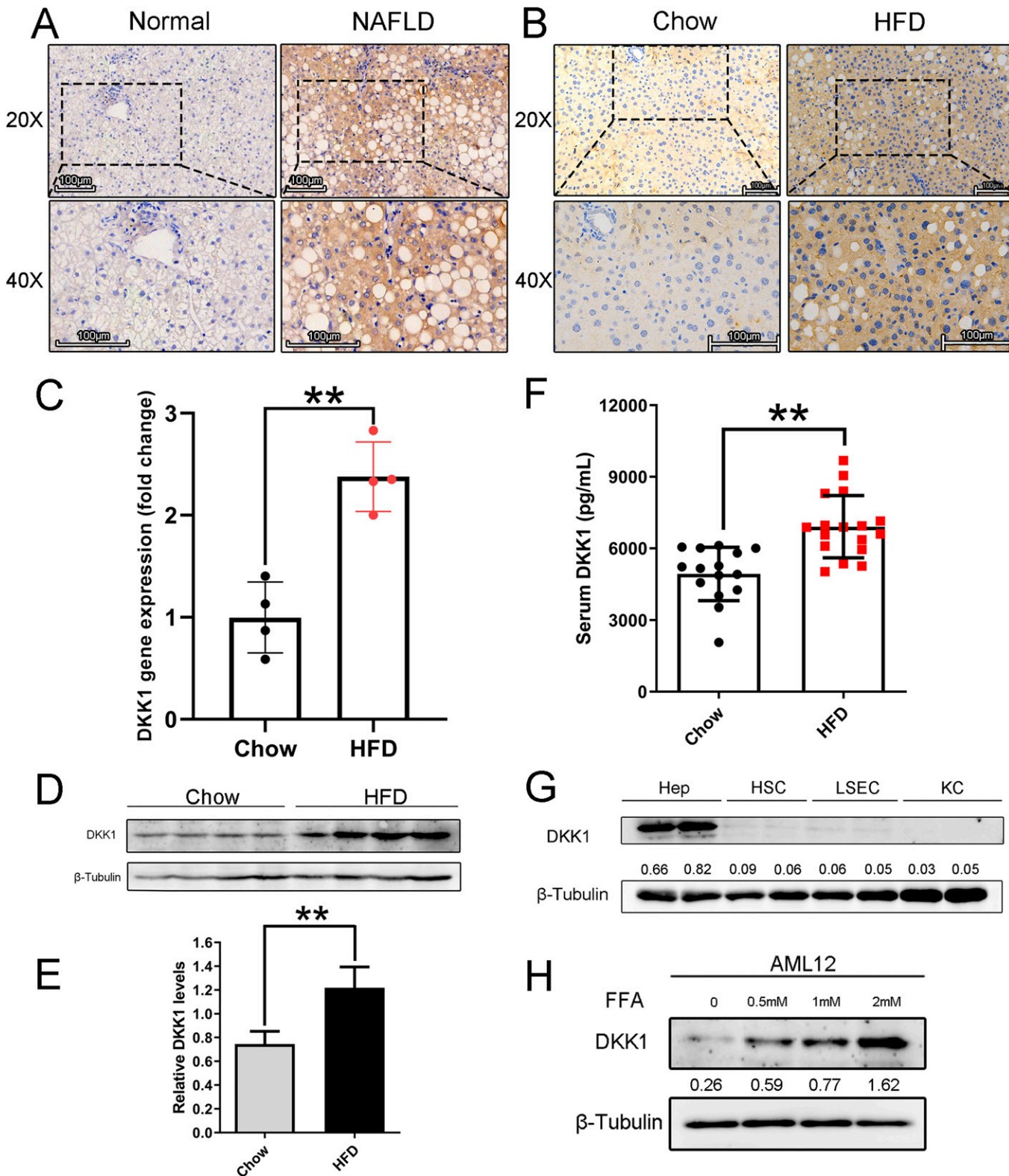

**Figure 1. DKK1 expression is elevated in livers of mice and patients with NAFLD.**
**(A)** Immunohistochemistry staining of DKK1 in the livers of patients with NAFLD (n = 5) and normal individuals (n = 3); scale bar = 100 $\mu m$, **(B)** in liver samples from WT mice fed with chow and HFD for 24 wk (n = 5). Scale bar = 100 $\mu m$. **(C)** qPCR analysis of DKK1 mRNA expression in livers of chow- and HFD-fed 24-wk (n = 4 mice for each group). **(D, E)** Representative Western blot of DKK1 in liver samples of chow- and HFD-fed mice (n = 4). Each lane represents liver lysates from individual mouse. **(F)** Serum DKK1 protein levels in chow- or HFD-fed mice for 24 wk (n = 15–17). **(G)** Western blot of DKK1 in different primary cells isolated from HFD-fed mice liver. Hep, hepatocytes;

and GFP fluorescence intensity measurements (Fig S2I) confirmed the status of OE or knock-down, respectively. Based on the in vitro validated data, the best efficacy of shRNA targeting sequences was chosen for the mice knock-down experiments.

The cellular TG contents in DKK1$^{-/-}$ AML12 (Fig 3B) and DKK1$^{-/-}$ HepG2 (Fig 3C) cells treated with FFA were markedly decreased (20.7% and 36.5%, respectively) compared with the DKK1$^{+/+}$ cells. Similarly, ORO staining showed more lipid droplets in FFA-treated DKK1$^{+/+}$ AML12 cells than in DKK1$^{-/-}$ AML12 cells (Fig 3A). In addition, a DKK1 inhibitor, WAY262611 (5 µM) was used for confirmation; as expected, the cellular TG contents in WAY262611-treated cells (Fig 3D and E) were significantly reduced (10.9% and 48.8%, respectively) compared with control cells. However, in LV-OE-DKK1 AML12 (Fig 3G) and HepG2 (Fig 3H) cells treated with FFA, cellular TG contents were markedly increased (42.9% and 180%, respectively) compared with those of the LV-GFP-NC controls; and those results were double confirmed with ORO staining (Fig 3F). To test whether DKK1 has direct effect on steatosis, the recombinant DKK1 (rDKK1) proteins (100 ng/ml) were administrated in FFA-induced cell steatosis models (Fig 3I and J). As expected, the cellular TG contents in rDKK1-treated cells were significantly increased compared with those of control cells (increased 57.6% in AML12 and 13.02% in HepG2).

### The analyses of DKK1-affected differential expressions of lipid metabolic genes

To investigate underlying mechanisms involved in the regulatory role of DKK1, gene expression profiles of the FFA-treated LV-GFP-NC and LV-OE-DKK1 AML12 cells were analyzed by RNA-seq. Compared with the LV-GFP-NC group, 84 genes significantly up-regulated and 62 genes significantly down-regulated in the LV-OE-DKK1 group (Fig 4A). Furthermore, KEGG pathway enrichment analysis revealed that the differentially expressed genes were significantly enriched in fat absorption among lipid metabolic pathways. In addition, the PI3K-Akt signaling pathway appeared in top enriched signaling pathways (Fig 4B). To further confirm the transcriptional effects of DKK1 on these genes, the expression of lipid metabolism–related genes in DKK1 OE and knockout AML12, HepG2 cells treated with FFA for 24 h was examined by qPCR. The fatty acid uptake–related gene CD36 appeared to be consistent significantly altered in different cell models, indicating that CD36 is a potential key gene downstream of DKK1 (Fig 4C–F). In addition, fatty acid synthesis, uptake and cholesterol synthesis–related genes were significantly changed, but the changes were not as consistent as the changes of CD36 in the four different tested cell lines. In comparison, little change in fatty acid oxidation genes was observed.

### CD36 was a key mediator of DKK1-driven steatosis, mediated by ERK-PPARγ signaling

Based on the change in CD36 mRNA induced by DKK1, we further analyzed the mRNA (Fig 2I) and protein levels of CD36 in animal models and cell lines with different DKK1 expression conditions and

confirmed that CD36 was elevated in the OE-DKK1 mice liver, whereas decreased in the sh-DKK1 mice liver compared with the one in GFP-NC control group (Fig 5A). Moreover, the protein expressions related to cholesterol synthesis, lipogenesis, and fatty acid oxidation were found with no significant changes in vivo and in vitro (Fig 5A and B).

To test the role of CD36 in DKK1-driven hepatic steatosis, the CD36 knockout HepG2 and AML12 cell lines were generated by the CRISPR/Cas9 editing, and the obtained knockout lines were validated by gene sequencing (Fig S2E and F) and Western blot (Fig S2G and H). Administration of rDKK1 (r-hDKK1 or r-mDKK1, 100 ng/ml) failed to increase FFA-induced steatosis of CD36 knockout hepatocytes; the ORO staining showed no significant differences either in CD36$^{-/-}$ HepG2 (Fig 5C) or CD36$^{-/-}$ AML12 cells (Fig 5D). Furthermore, the measurement of cellular TG contents confirmed the scenario (Fig 5E and F). Conversely, rDKK1 administration significantly increased TG 44.1% (P < 0.001) in WT HepG2 and 15.3% (P < 0.005) in AML12. It suggests that the DKK1 promoted steatosis is CD36 dependent.

To investigate the molecular basis of DKK1 regulating CD36 transcription, the CD36 gene promoter luciferase reporter system was constructed and tested to determine whether DKK1 increased CD36 transcriptional activity. As shown in Fig 5G, OE of DKK1 significantly increased the luciferase activity of the CD36 promoter (increased 170%, P < 0.001). CD36 is a target of PPARγ (Zhou et al, 2008), and it is suggested that ERK signaling activates CD36 expression through PPARγ in hepatic steatosis (Zhang et al, 2019). The mRNA level of PPARγ was significantly increased in DKK1 OE cells (Fig 4C), whereas markedly diminished in DKK1 knockout cells (Fig 4E and F). Furthermore, the phosphorylation of ERK and nucleus PPARγ was significantly increased in OE-DKK1 AML12 compared with the control (Fig 5H).

### DKK1-induced insulin resistance is related to the phosphorylation of JNK-AKT-FOXO1

As DKK1 is a natural antagonist of the Wnt signaling (Gonzalez-Sancho et al, 2005; Menezes et al, 2012) and the Wnt pathway is reportedly involved in metabolic syndromes (Ackers & Malgor, 2018; Abou Ziki & Mani, 2019) such as NAFLD (Liu et al, 2011; Wang et al, 2015), the expressions of key genes in Wnt signaling were examined in DKK1 knockout HepG2 cells treated with FFA. We found that the mRNA expressions of ROCK1, JNK, and PPARγ were markedly decreased in DKK1-KO HepG2 cells compared with the WT HepG2 cells (Fig S3B), and the protein expression of phosphorylated JNK was decreased as in DKK1-KO HepG2 cells (Fig S3C).

Activation of JNK is crucial to the development of insulin resistance and steatosis (Hirosumi et al, 2002; Czaja, 2010). To further evaluate the effect of DKK1 on glucose tolerance and insulin sensitivity, glucose tolerance test (GTT) and insulin tolerance test (ITT) were performed in OE-DKK1, sh-DKK1, and GFP-NC mice. As expected, the HFD-induced insulin resistance in sh-DKK1 mice was significantly decreased, whereas the GTT was markedly increased in

---

HSC, hepatic stellate cells; LSEC, liver sinusoidal endothelial cells; KC, Kupffer cells. **(H)** Western blot analysis of DKK1 in AML12 cells under different concentrations of FFA exposures (N = 2). **P < 0.01 as compared with the indicated controls by two-tailed t tests (two groups). All data are shown as the means ± SD. Source data are available for this figure.

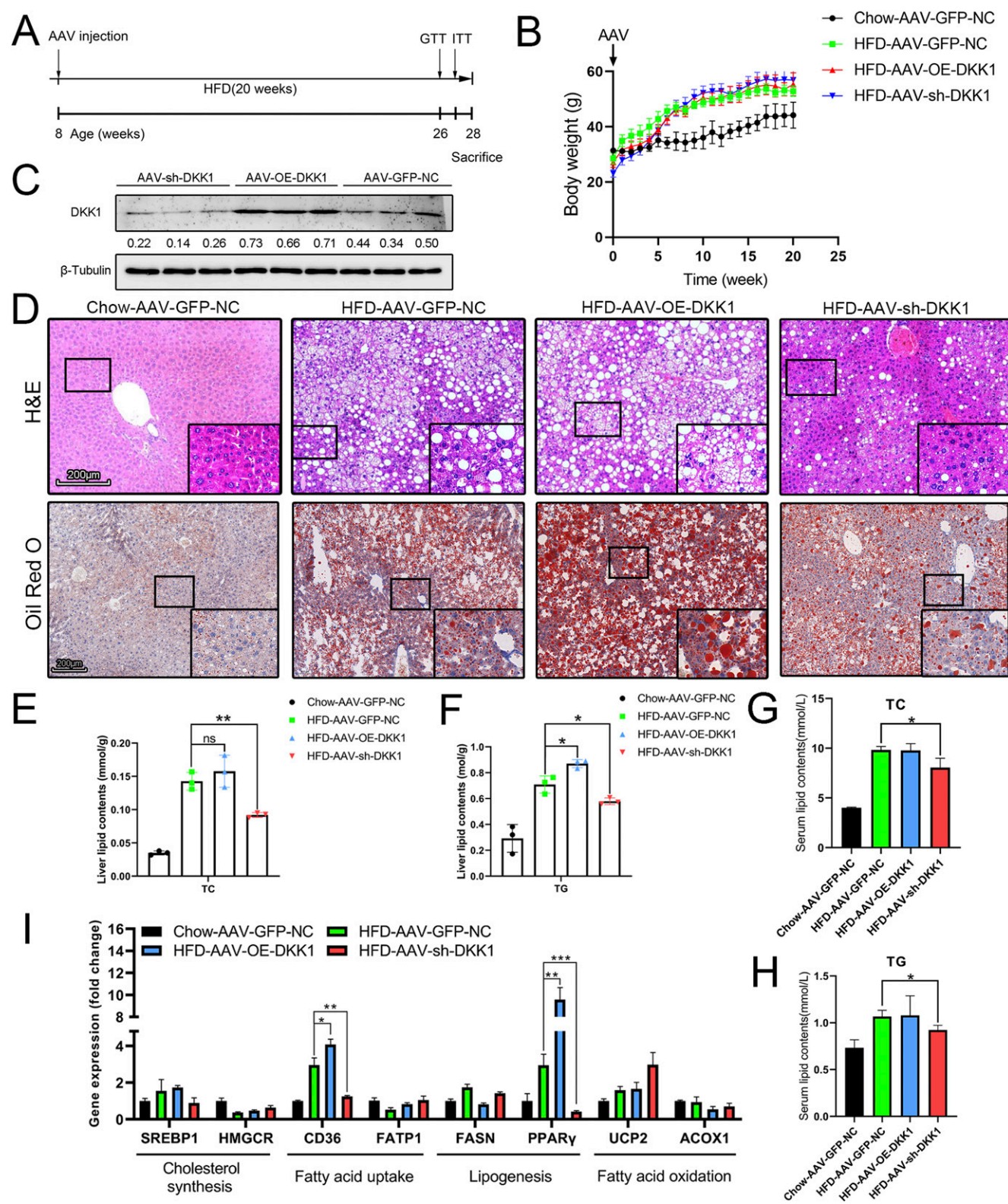

**Figure 2. Liver-specific DKK1 overexpression exacerbates HFD-induced liver steatosis.**
**(A)** Schematic illustration of experiment procedure. AAV tail intravenous injection with AAV-GFP-NC (n = 4), AAV-OE-DKK1 (n = 5), or AAV-sh-DKK1 (n = 6) and then fed HFD for 20 wk before euthanasia. **(B)** Dynamic body weight tracking of chow- and HFD-fed mice with DKK1 manipulations. **(C)** Representative Western blot of DKK1 in liver samples of HFD-fed mice after 20 wk (n = 3). Each lane represents liver lysates from individual mouse. **(D)** H&E and ORO staining. Scale bar = 200 $\mu$m; **(E, F)** lipid contents in liver of mice with different DKK1 gene manipulations under chow or HFD fed (n = 3). **(G, H)** Serum lipid contents with different DKK1 gene manipulations under chow or

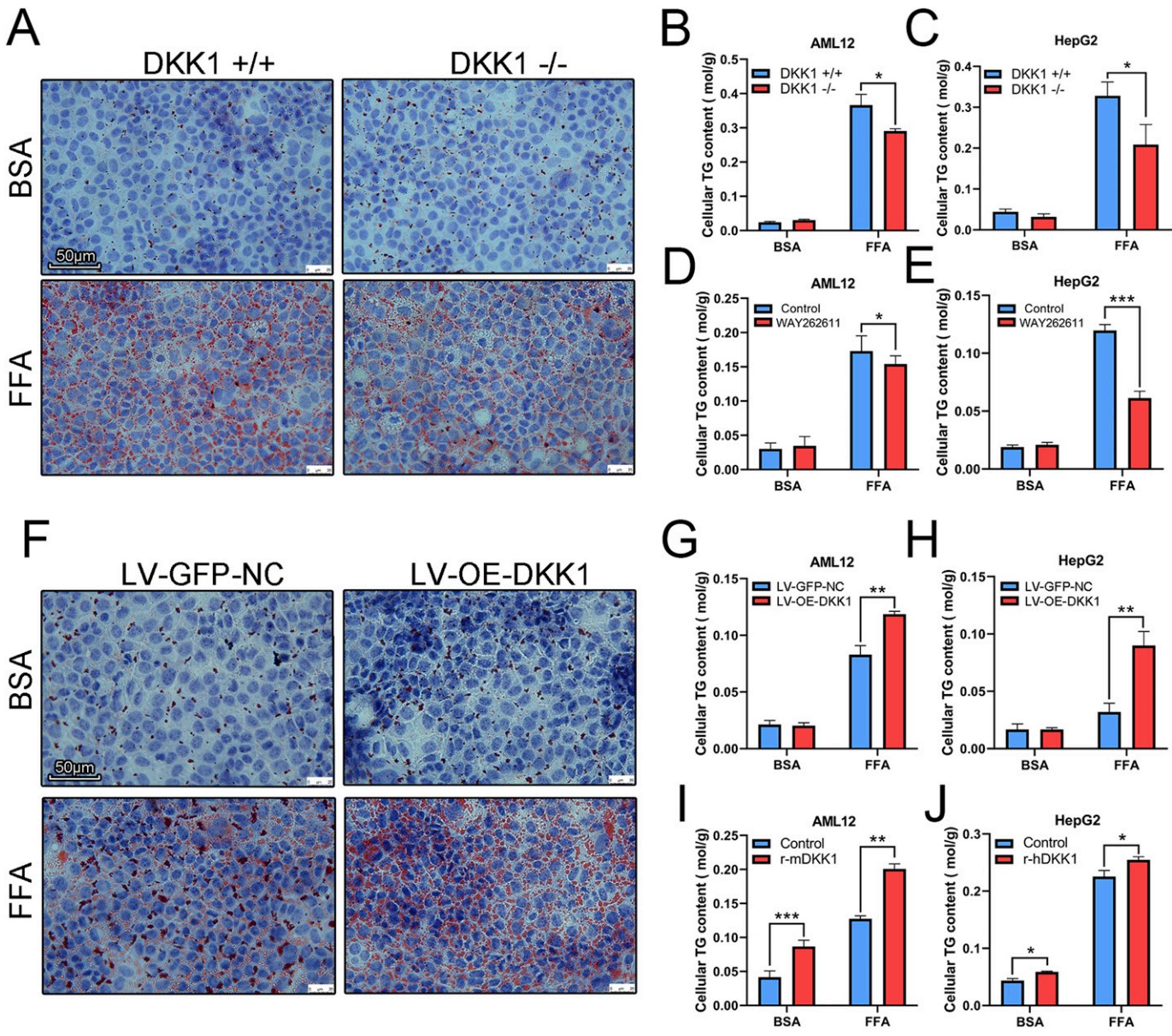

**Figure 3. Manipulations of DKK1 expression confirmed the linkage of DKK1 and steatosis in cell lines.**
**(A, B, C)** Under DKK1 knockout conditions, cell steatosis was induced by FFA (A, B, C), n = 3 in each group. **(A, B, C)** Oil red O staining (A) scale bar = 50 μm; TG measurements in DKK1$^{-/-}$ AML12 (B) and DKK1$^{-/-}$ HepG2 (C) cells. n = 3 in each group. **(D, E)** The changed TG was further confirmed with administration of a DKK1 inhibitor, WAY262611 in AML12 (D) and HepG2 (E) cells. n = 3 in each group. **(F, G, H)** On other hand, under DKK1 overexpression condition, the induced steatosis status was parallel analyzed, oil red O staining (F), and TG measurements in DKK1$^{-/-}$ AML12 (G) and DKK1$^{-/-}$ HepG2 (H) cells. n = 3 in each group. **(I, J)** Furthermore, the changed trend of TG was further confirmed with administration of recombinant DKK1 protein in AML12 (I) and HepG2 (J) cells. n = 3 in each group. *P < 0.05, **P < 0.01, ***P < 0.001 as compared with the indicated controls by two-tailed *t* tests. All data are shown as the means ± SD.
Source data are available for this figure.

OE-DKK1 mice compared with GFP-NC controls (Fig 6A and B). To further identify the relationship between the DKK1 and JNK pathway, rDKK1 protein and FFA were used to incubate in AML12 cells. The phosphorylation of JNK was significantly increased upon rDKK1 stimulation (Fig 6C). Because insulin resistance is associated with PI3K–AKT-FOXO1 axis (Tilg & Moschen, 2008; Zhang et al, 2012), the phosphorylated AKT and FOXO1 were examined, and it was found that DKK1 decreased phosphorylation of AKT (Ser473) and FOXO1 (Fig 6D).

HFD fed. **(I)** The expression confirmations of lipid metabolism–related genes in liver of mice with different DKK1 gene manipulations under chow or HFD fed (n = 3). *P < 0.05, **P < 0.01, ***P < 0.001 as compared with the indicated controls by two-tailed *t* tests. All data are shown as the means ± SD. ns, not significant.
Source data are available for this figure.

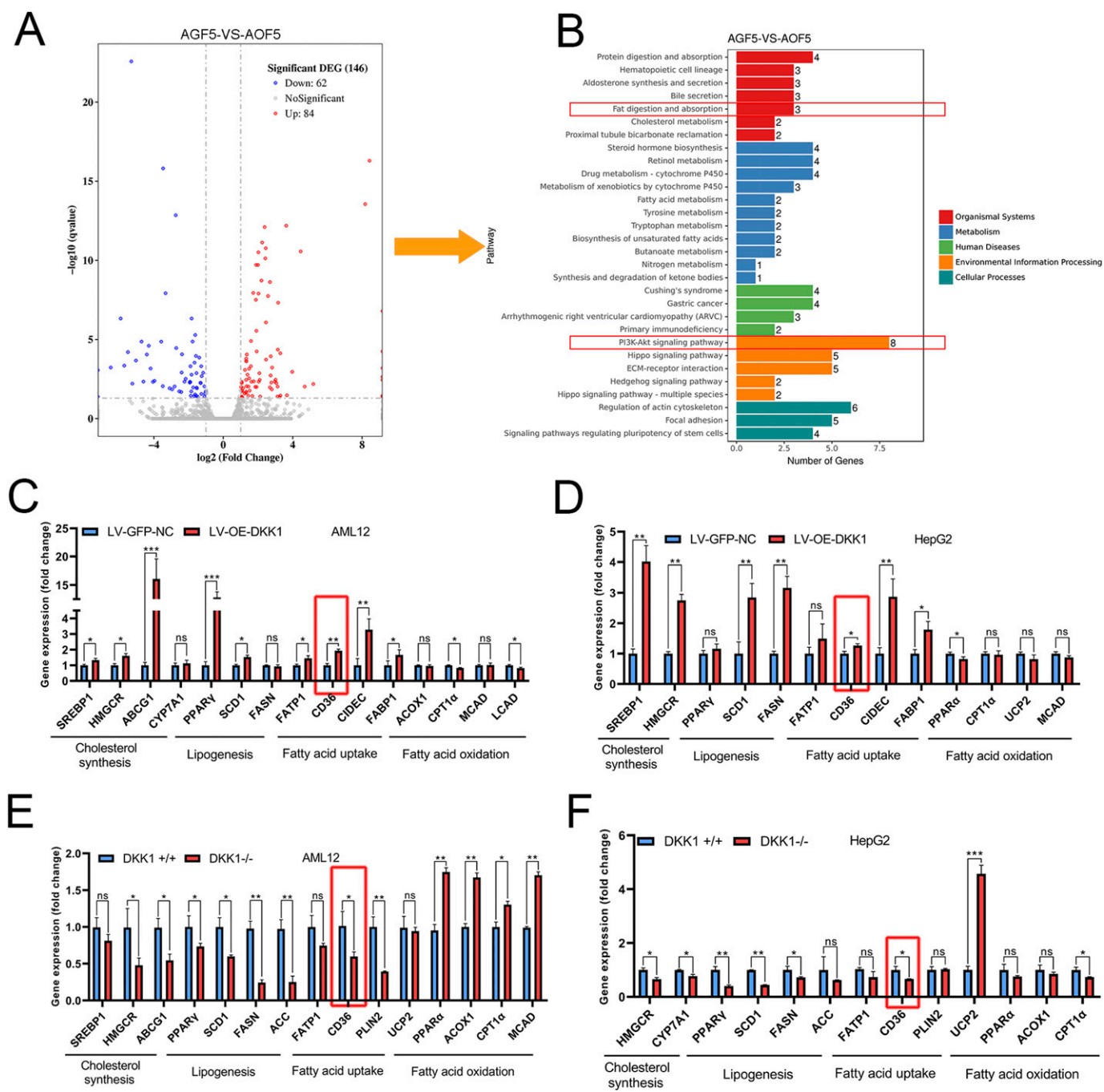

**Figure 4. Gene expression profiling analyses and confirmation of DKK1-affected expressions of lipid metabolic genes.**
**(A, B)** Volcano map (A) and KEGG analysis (B) of up- and down-regulated genes in LV-OE-DKK1 AML12 cells compared with LV-GFP-NC AML12 cells under FFA exposure for 24 h. **(C, D, E, F)** The expression confirmations of lipid metabolism–related genes in cell lines with different DKK1 gene manipulations under FFA treatment; DKK1 overexpression in AML12 (C) and HepG2 (D); DKK1 knockout AML12 (E) and HepG2 (F) cells. n = 3 in each group. *P < 0.05, **P < 0.01, ***P < 0.001 as compared with the indicated controls by two-tailed t tests. All data are shown as the means ± SD. ns, not significant.
Source data are available for this figure.

## Discussion

Using liver-specific DKK1 OE or knock-down mice, the evidence gathered from both pros and cons indicated that hepatocyte DKK1 is a key player in HFD-induced NAFLD initiation and progression.

Continuous challenge of the HFD progressively up-regulated hepatic DKK1 that activated the ERK-PPARγ-CD36 pathway to promote fatty acid uptake of hepatocytes. On the other hand, elevated DKK1 expression activates the JNK signaling pathway, which further inhibits AKT-FOXO1 phosphorylation cascades, promoting insulin

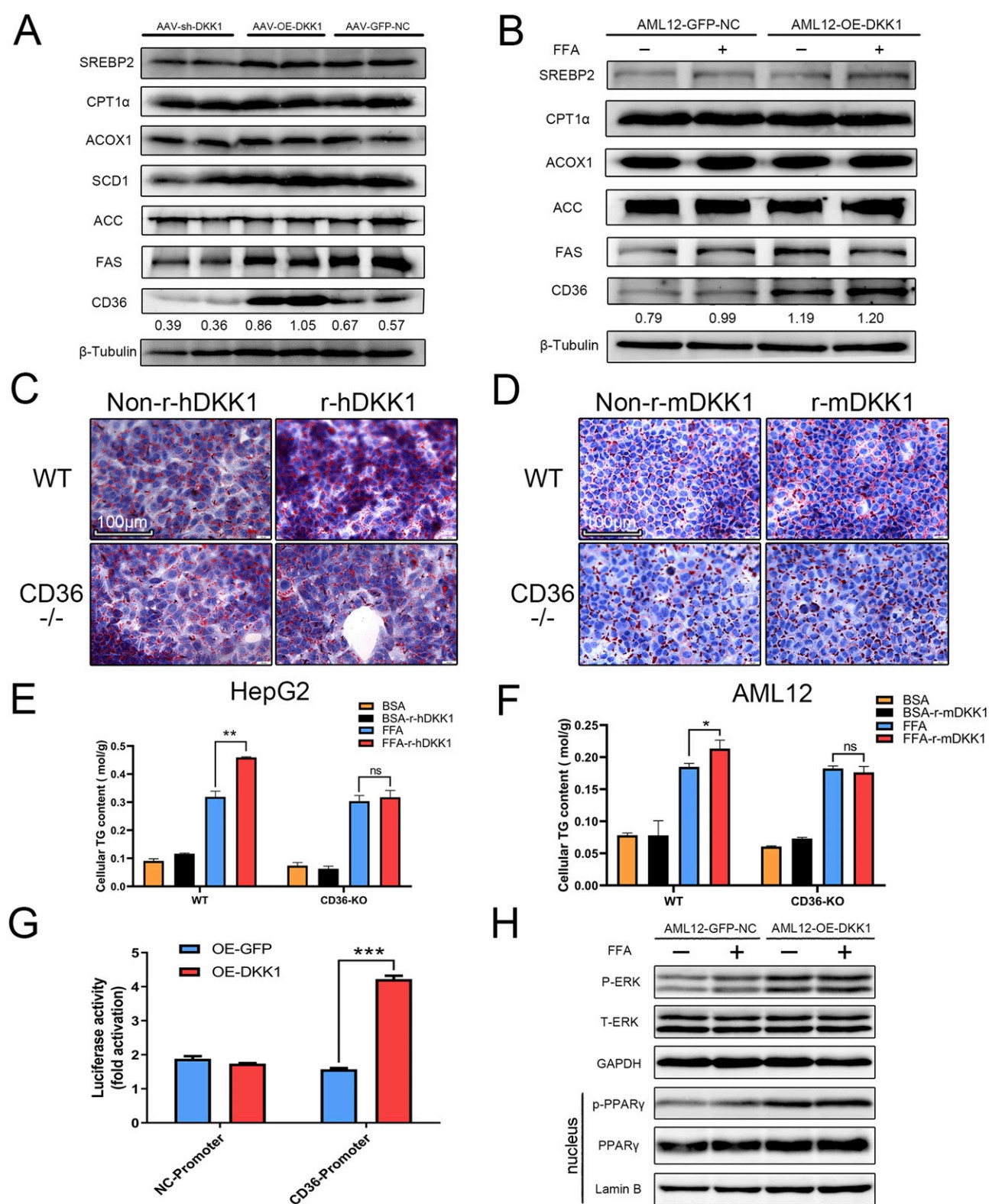

**Figure 5. CD36 is a key mediator of DKK1-driven steatosis.**
**(A, B)** Representative Western blot analyses of lipid metabolism–related proteins in DKK1 overexpression- or knock-down mice (A) and DKK1 overexpression in AML12 cell line (B), in which the numbers marked above the controls were the ratio between CD36 compared with the $\beta$-Tubulin. **(C, D, E, F)** The ORO staining of WT and CD36$^{-/-}$ cells cultured with recombinant DKK1 protein (r-hDKK1 or r-mDKK1) with or without FFA induction (C, D) scale bar = 100 $\mu$m, and measurement of TG contents in CD36$^{-/-}$ HepG2 (E) and CD36$^{-/-}$ AML12 (F) cells. n = 3 in each group. **(G)** The CD36-promoter-driven luciferase reporter assay under DKK1 overexpression condition. n = 3 in each

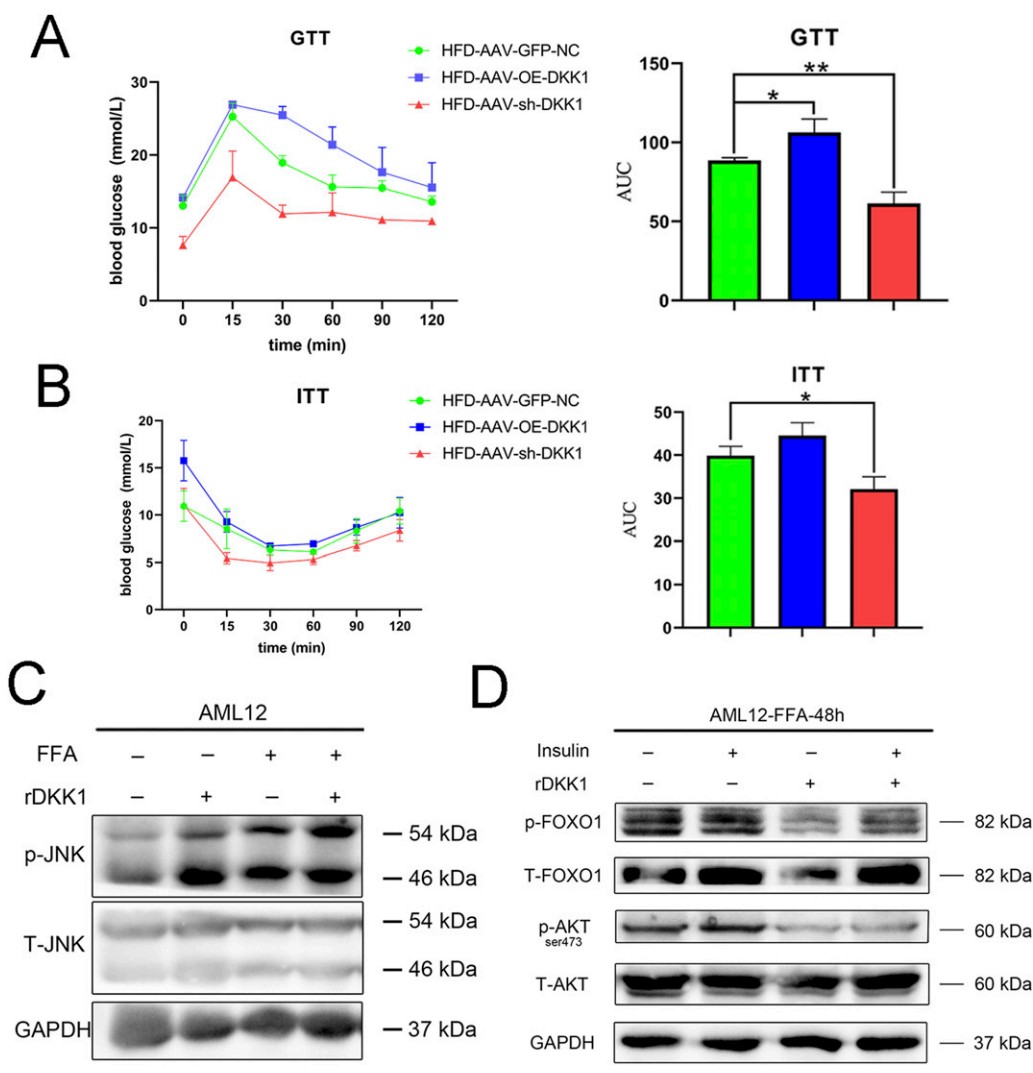

**Figure 6. DKK1-induced insulin resistance in mice which was related to the phosphorylation of JNK-AKT-FOXO1.**
**(A, B)** The GTT (A) and ITT (B) analyses of mice with treatments of AAV-GFP-NC, AAV-OE-DKK1, or AAV-sh-DKK1 and HFD-fed 20 wk (n = 3). **(C)** Western blot analyses of phosphorylated JNK in AML12 cells under rDKK1 stimulation (N = 2). **(D)** Representative Western blot analyses of phosphorylated AKT and FOXO1 in AML12 cells under insulin and rDKK1 stimulation (N = 2). *$P < 0.05$, **$P < 0.01$, ***$P < 0.001$ as compared with the indicated controls by two-tailed $t$ tests. All data are shown as the means ± SD. Source data are available for this figure.

resistance and glucose metabolism disturbances, thereby accelerating excessive lipid accumulation and steatosis (Fig 7).

To delineate the pathologic role of DKK1 in steatosis, first, we found that the hepatic DKK1 expression status was progressively up-regulated as the NAFLD condition advanced in clinical patient and NAFLD mice liver samples and that was confirmed by in vitro fatty acid–induced hepatocyte steatosis. Notably, serum DKK1 expression was elevated in NAFLD mice, consistent with the previous results in NAFLD patients (Polyzos et al, 2016). In type 2 diabetes (Lattanzio et al, 2014) and atherosclerosis (Ueland et al, 2009), serum DKK1 expression was found to be elevated and speculated

that DKK1 was mainly derived from platelet activation. However, this speculation remains subject to further experimental verification to weight the serum DKK1 level contribution from the hepatic and platelet resources.

Furthermore, liver-specific OE or knock-down of DKK1 in HFD mice and gene manipulations in hepatocyte cell lines revealed that up-regulated DKK1 aggravates hepatic steatosis and insulin resistance. However, the regulation of lipid metabolism by DKK1 seemed to be diversified and paradoxical in previous reports. A decrease of DKK1 contributes to placental lipid accumulation in an obesity-prone rat model (Strakovsky & Pan, 2012), whereas

group. **(H)** Western blot analysis shows the overexpression of DKK1 increased pERK and nucleus PPARγ in AML12 cells with independent duplicates. *$P < 0.05$, **$P < 0.01$, ***$P < 0.001$ as compared with the indicated controls by two-tailed $t$ tests. All data are shown as the means ± SD. Source data are available for this figure.

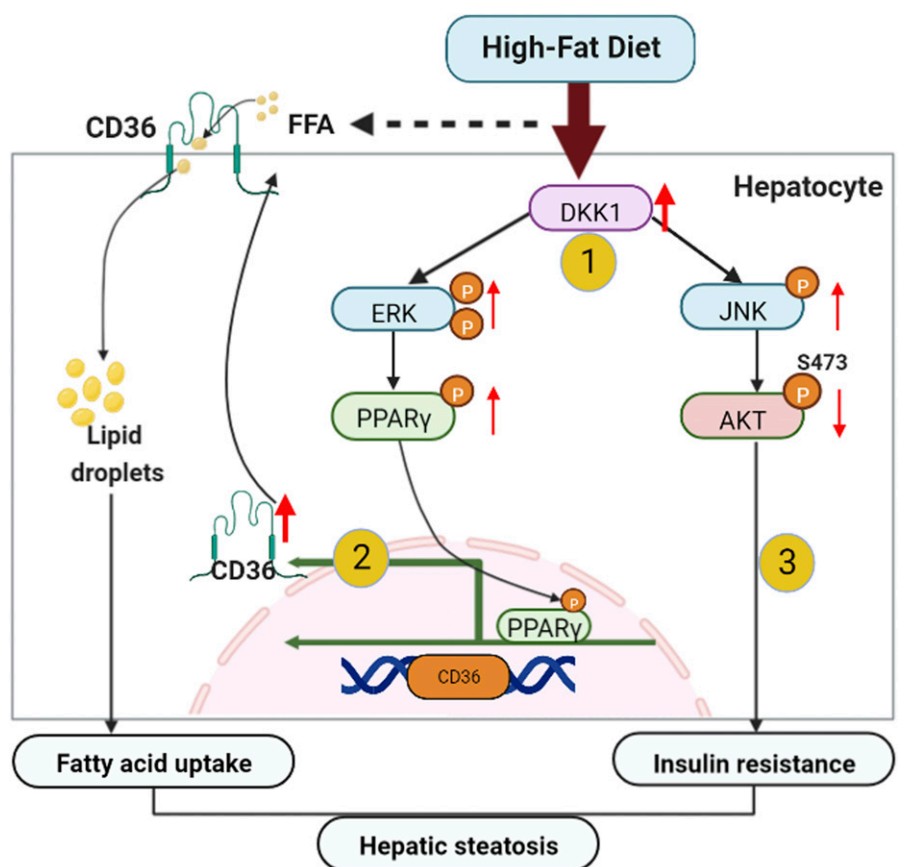

**Figure 7. Schematic illustration of cellular and molecular events underlying hepatocyte DKK1-regulated hepatic steatosis.**
In response to continuous challenge with HFD, (1) the DKK1 expression is mainly induced in hepatocytes and the increased serum DKK1 could have served as a diagnostic bio-marker for steatohepatitis progression. (2) DKK1 enhances hepatic CD36 expression by activating ERK-PPARγ signaling and consequently leads to increased hepatic fatty acid uptake and hepatocyte steatosis. (3) DKK1 activates JNK to decrease the phosphorylation of AKT and FOXO1, leading to insulin resistance. Hepatic fatty acid uptake and insulin resistance synergistically exacerbate fatty acid accumulation and the resultant hepatic steatosis.

inhibiting DKK1 by siRNA decreased lipid accumulation of adipocytes in human hypertrophic obesity (Gao et al, 2017). We speculate that these phenotypic differences may be related to the different cell types and pathological conditions.

In this study, we primarily focused on the role of DKK1 in NAFLD development, and our findings are consistent with the previous findings where DKK1 was identified as a risk factor (Li et al, 2016) in cardiovascular disease. Interestingly, DKK3 was reported to be involved in inhibition of hepatic steatosis (Xie et al, 2016), prevention of myocardial infarction (Bao et al, 2015), and anti-atherosclerosis (Yu et al, 2017). This may be because of differences in DKK member's interaction behaviors; DKK1, DKK2, and DKK4 regulate the Wnt signaling through binding the same effectors (Niehrs, 2006), but not DKK3.

Based on the suggestion of gene expression profiling, we focused on screening and verification of those DKK1-affected lipid metabolism genes and found that both mRNA and protein levels of CD36 were altered and correlated with the gain or loss of expression of DKK1. CD36 is a fatty acid translocase mediating the FFA uptake and expression in platelets, adipocytes, macrophages, vascular endothelial cells and hepatocytes (Silverstein & Febbraio, 2009; Pietka et al, 2014). Hepatic CD36 up-regulation has been significantly associated with insulin resistance and steatosis in NAFLD patients (Koonen et al, 2007; Heebøll et al, 2017). Hepatic CD36 is mainly regulated by nuclear receptors LXR, PXR and PPARγ in promoting steatosis (Zhou et al, 2008). We observed that the PPARγ expression was elevated in DKK1 overexpressing AML12 cells, suggesting that

DKK1 may potentially regulate CD36 through PPARγ pathway. This speculation is supported by other evidence where DKK1 up-regulates PPARγ expression during adipogenesis (Lu et al, 2016) and osteoprotegerin-mediated up-regulation of CD36 through ERK-PPARγ (Zhang et al, 2019). Indeed, our CD36 promoter luciferase reporter assay and ERK or PPARγ western results demonstrated that DKK1 activates CD36 expression through ERK-PPARγ signaling.

Abnormal activation of the non-canonical Wnt signaling pathway is an important factor leading to NAFLD (Wang et al, 2015). We examined the status of those non-classical Wnt factors in DKK1-overexpressing hepatocytes and found that the phosphorylated JNK was significantly increased. It is consistent with previous findings that OE of DKK1 in both mesothelioma cell lines (Lee et al, 2004) and zebrafish (Caneparo et al, 2007) activates the non-canonical Wnt signaling pathway and higher phosphorylated JNK. Notably, JNK was aberrantly activated in patients with NAFLD, and this aberrant activation was accompanied with symptoms such as insulin resistance and inflammation (Hirosumi et al, 2002; Czaja, 2010). DKK1-overexpressed mice also showed significant insulin resistance, suggesting that DKK1 may cause insulin resistance through activation of the JNK signaling and thus exacerbate liver steatosis. We further examined the effect of rDKK1 protein on hepatocyte insulin sensitivity and found that DKK1 reduced the phosphorylation levels of key factors in the insulin pathway that is in agreement with other study (Ling et al, 2013). Surprisingly, increased CD36 has also being linked with insulin resistance through

JNK (Karunakaran et al, 2021). Loss of CD36 impairs hepatic insulin signaling in a low-fat diet fed mice (Yang et al, 2020) but improves insulin sensitivity in HFD-fed mice (Wilson et al, 2016). Obviously, it needed further validation to clarify the detailed role of CD36 in DKK1-induced insulin resistance.

In summary, we found that the increased DKK1 is a key factor for hepatic steatosis in HFD-induced NAFLD mice model, which mainly relies on the activation of ERK-PPARγ-CD36 signaling to increase fatty acid uptake and steatosis; on the other hand, DKK1 activates the JNK phosphorylation, resulting in insulin resistance. Therefore, DKK1 may be a novel target for drug development for NAFLD and metabolism disorders.

# Materials and Methods

### Mice and treatment

This animal study was approved by the Institutional Animal Care and Use Committee of the Guangzhou Institutes of Biomedicine and Health, Chinese Academy of Sciences. All animal feeding, experimental handling, and surgical operations were in accordance with animal ethics and welfare requirements. 6–8-wk male C57BL/6 mice were purchased from the Vital River Laboratory Animal Technology Co. Ltd. The mice were housed three to five per cage in an SPF animal house at the Experimental Animal Center of the Guangzhou Institutes of Biomedicine and Health, Chinese Academy of Sciences. Mice were kept in a standard environment (12-h daylight cycle), with free access to food and water. The NAFLD mice model was established by feeding an HFD (60% fat, 5.21 kcal/g, D12492; Research Diet Inc.) continuously for 20 or 24 wk. The control diet was cobalt-60-irradiated normal chow diet purchased from the Guangdong Medical Laboratory Animal Center. The body weight was measured weekly throughout the treatment period.

To generate hepatocyte-specific DKK1 knock-down (sh-DKK1) and OE (OE-DKK1) mice, we transduced an AAV system (designed and synthesized by Hanbio) carrying AAV-TBG-sh-DKK1 and AAV-TBG-OE-DKK1 (titer: $1 × 10^{12}$ vg ml$^{-1}$) into mice at a dose of $2 × 10^{10}$ vg per mouse through tail vein injection. The AAV-TBG-GFP was transduced as a negative control. Detailed DKK1 shRNA oligonucleotide sequences were used as follows: CCGGGCTGCATGAGGCACGC-TATGTTTCAAGAGAACATAGCGTGCCTCATGC
AGCTTTTTGAATT. The OE cDNA sequences are mouse Dkk1 (Gene ID: 13380; NM_010051.3).

### Human liver samples

All human liver samples used in this study were collected from the First Affiliated Hospital of Jinan University, and donor consent was obtained. All review procedures involving human samples were approved by the Review Committee of the First Affiliated Hospital of Jinan University and were in accordance with the guidelines outlined in the Declaration of Helsinki. Fatty liver samples were obtained from NASH patients who had undergone liver biopsy or liver transplantation. Non-steatosis liver samples were obtained from donor livers that were unsuitable for transplantation for non-

hepatic reasons, and NAFLD patients were rated according to the "NASH Clinical Research Network" scoring system with five cases per group.

### Histological analyses

Liver samples were fixed overnight in 4% paraformaldehyde and embedded in paraffin. Paraffin-embedded liver samples were sectioned at 4 μm for H&E staining and immunohistochemistry. After being deparaffinized, rehydrated, and washed in PBS, sections were stained with standard hematoxylin for 6 min and aqueous eosin for 4 min to perform H&E staining. Finally, the morphologic characteristics of the liver were visualized after sections were dehydrated, transparent, and sealed. To perform immunohistochemistry in liver sections, samples were unmasked with incubated citrate-EDTA buffer at high fire in a microwave oven for about 5 min. Sections were incubated in 3% hydrogen peroxide for 10 min to block endogenous peroxidase, followed by incubation in 10% FBS for 60 min at room temperature to block non-specific binding and then incubated with primary antibodies. DKK1 antibody (ab93017, 1: 1,000) was incubated at 4°C overnight. HRP-labeled goat anti-rabbit (KC-RB-035; Aksomics) was used as the secondary antibody. 3, 3′-DAB (K3468; Dako) was employed for the detection procedure. To perform tissue ORO staining, liver tissues were embedded in Tissue-Tek OCT compound (4583; Sakura Finetek). The frozen liver sections were stained with 0.2% ORO and counterstained with hematoxylin to visualize lipid droplets in the liver.

### Serum biochemical analyses and ELISA

Cardiac blood was obtained after fasting overnight, and plasma was isolated via centrifugation at 3,552$g$ for 15 min after the blood was left to stand for 2 h at room temperature. ALT, AST, TC, TG levels were measured using an automatic biochemical analyzer (7020; Hitachi) in Fengrui Biotechnology Co. Serum levels of DKK1 were measured using a commercially available mouse ELISA kit (EK0925; Boster Biological) according to the manufacturer's instructions.

### Mouse metabolic assays

Blood was obtained from the tail veins of living mice. To perform the GTTs and ITTs, 2 g/kg glucose (Sigma-Aldrich) was orally administrated to mice, and 1 U/kg insulin (Novolin R; Novo Nordisk) was i.p. injected to mice. Blood glucose was determined using a glucometer (ACCU-CHEK Performa; Roche) before glucose oral administration (after a 12 h fast) or insulin injection (after a 6 h fast) and 15, 30, 60, 90, 120 min after injection. GTT and ITT were performed 2 and 1 wk before euthanasia, respectively. The area under the curve was calculated for the GTT and ITT with GraphPad Prism program.

### Cell lines and cell culture

All cell lines used in this study were maintained at 37°C and 5% $CO_2$ with humidified air. 293T (CRL-11268; ATCC) cells were cultured in DMEM (high glucose; Gibco) supplemented with 10% FBS, and HepG2 (HB-8065; ATCC) cells were cultured in DMEM (low glucose; Gibco) supplemented with 10% FBS. AML12 (CRL-2254; ATCC) cells

were cultured in DMEM/F12 supplemented with 10% FBS, 1% ITS (10 μg/ml insulin, 5.5 μg/ml transferrin, and 5 ng/ml selenium), and 40 ng/ml dexamethasone. Mycoplasma contamination was checked once in 2 mo.

CRISPR/Cas9-induced gene knockout in AML12 and HepG2 cells was established as previously described (Yang et al, 2021) that includes the following steps: design sgRNAs, synthesize and ligate gRNA sequence to p×459v2 cloning vector, transfect with 2 μg of each sgRNA plasmid using Lipofectamine 3000 in 0.8 million AML12 and HepG2 cells, then seed onto 24-well plates, Targeted cells were selected with 100 μg/ml puro for 2 d after 24 h recovery. Then the puromycin-resistant cells were re-plated on 96-well plates for single-cell culture; PCR was used for sequencing identification. Primers for PCR amplification and plasmid constructs are shown in Table 1.

Lentiviral vectors were constructed to produce lentiviruses OE of DKK1 in AML12 and HepG2 cells. Mouse and human DKK1 cDNA sequences were subcloned into pCDH-CMV-MCS-EF1-copGFP-T2A-Puro vector between by Guangzhou IGE Biotechnology Co., Ltd. Then 293T cells were co-transfected with packaging plasmid psPAX2 (Addgene), envelope plasmid pMD2.G (Addgene), and respective OE vectors for the production of lentiviruses. The empty plasmid was transduced as a negative control. The lentivirus particles were harvested and purified 48 h after transfection, redissolved in sterile PBS, aliquoted, and stored at 80°C. Purified lentiviruses were infected with AML12 and HepG2 cells and the negative cells that are not infected with overexpressing lentivirus were screened out by puromycin. Green fluorescence density and protein expression examined by Western blot analysis were used to validate the successful construction of DKK1 OE cell lines.

### Intracellular TG levels

24 h after treatment with a mixture of FFA (palmitate and oleate, 1:2; Sigma-Aldrich) at the concentration of 0.5 mM (Feldstein et al, 2004), cultured AML12 and HepG2 cells were lysed at room temperature for 10 min and collected by centrifugation at 2,664$g$ for 6 min. The intracellular TG levels were detected using a commercially available TG Assay Kit (Applygen Technologies) according to the manufacturer's protocol.

### Cell ORO staining

Cells were seeded in a 24-well plate with sterile slides placed in advance. After stimulation with FFA for 24 h, cells were washed three times with PBS before being fixed with 10% neutral formaldehyde for 15 min. After two washes in PBS, cells were stained with ORO using an oil red staining kit purchased from KeyGEN Biotechnology Co., Ltd, following the vendor's recommended protocols. At last, the slides were picked up and sealed with glycerin gelatin after being dyed with hematoxylin for 1 min. The stained lipid droplets were visualized and photographed under an inverted microscope (Leica).

### RNA sequencing and processing

Gene expression profiles were analyzed using OE-DKK1 and GFP-NC AML12 cells treated with 0.5 mM FFA for 24 h. The total RNA of each

**Table 1.  List of primers used for DKK1 and CD36 knock out and identify.**

| Primer name | Primer sequence (5′-3′) |
|---|---|
| L-E2-DKK1-gRNA-F | CACCGTCACGCTATGTGCTGCCCC |
| L-E2-DKK1-gRNA-R | AAACGGGGCAGCACATAGCGTGAC |
| R-E2-DKK1-gRNA-F | CACCGCGTTTTCGGCGCTTCCTGC |
| R-E2-DKK1-gRNA-R | AAACGCAGGAAGCGCCGAAAACGC |
| E2-DKK1-CHECK-F | GTACCCGGGCGGGAATAAG |
| E2-DKK1-CHECK-R | ATAGACGCTCAAAGGCTGGAC |
| L-E1-MDKK1-gRNA-F | CACCGATGATGGTTGTGTGTGCAG |
| L-E1-MDKK1-gRNA-R | AAACCTGCACACACAACCATCATC |
| R-E1-MDKK1-gRNA-F | CACCGAGAGCCATCATTGTAAACA |
| R-E1-MDKK1-gRNA-R | AAACTGTTTACAATGATGGCTCTC |
| E1-MDKK1-CHECK-F | TTGTTGTCTTCCCTGAGGAGC |
| E1-MDKK1-CHECK-R | ATCTTCAGCGCAAGGGTAGG |
| H-L-CD36-gRNA-F | CACCGAGATGGCACCATTGGGCTGC |
| H-L-CD36-gRNA-R | AAACGCAGCCCAATGGTGCCATCTC |
| H-R-CD36-gRNA-F | CACCGTTCACTATCAGTTGGAACAG |
| H-R-CD36-gRNA-R | AAACCTGTTCCAACTGATAGTGAAC |
| H-CD36-CHECK-F | GCATGCTACCATCTGCCGTA |
| H-CD36-CHECK-R | TTGCCCACTGGTACAGCTAC |
| M-L-CD36-gRNA-F | CACCGCCAAAACTGTCTGTACACAG |
| M-L-CD36-gRNA-R | AAACCTGTGTACAGACAGTTTTGGC |
| M-R-CD36-gRNA-F | CACCGTGTGCAAAACCCAGATGACG |
| M-R-CD36-gRNA-R | AAACCGTCATCTGGGTTTTGCACAC |
| M-CD36-CHECK-F | TTGACTAAGGGAGTGTTGCCA |
| M-CD36-CHECK-R | GGTCGACTAGGCCATCCTTT |

sample was isolated using TRI Reagent (MRC) according to the manufacturer's protocol. The quality and quantity of total RNA were assessed using an Agilent 2100 Bioanalyzer (Agilent Technologies). The RNA from three samples within each group was mixed in equal amounts. 1 μg total RNA from each group was sent to GENEWIZ Biological Technology Co., Ltd, for subsequent RNA sequencing and analysis. The sequencing data obtained from the RNA-Seq were submitted to the National Center for Biotechnology Information's GEO database under accession number GSE197746.

### qPCR

Total RNA was isolated with TRI Reagent (MRC) according to manufacturer's instructions, and 2 μg of total RNA was used for cDNA synthesis with a ReverTra Ace qPCR RT Master Mix Kit (FSQ-301; TOYOBO) according to the manufacturer's protocol. SYBR Green (YEASEN Biotech) and Bio-Rad CFX96 Real-Time System (Bio-Rad) were applied to quantify PCR amplification. The mRNA expression levels were calculated using the $2^{-\triangle\triangle Ct}$ method. The primer sequences are summarized in Table 2.

**Table 2.  List of specific primers sequence used for qPCR analysis.**

| Primer name | Primer sequence (5'-3') |
|---|---|
| H-SREBP1-QP-F | ACAGTGACTTCCCTGGCCTAT |
| H-SREBP1-QP-R | GCATGGACGGGTACATCTTCAA |
| H-HMGCR-QP-F | TGATTGACCTTTCCAGAGCAAG |
| H-HMGCR-QP-R | CTAAAATTGCCATTCCACGAGC |
| H-SCD-QP-F | TCTAGCTCCTATACCACCACCA |
| H-SCD-QP-R | TCGTCTCCAACTTATCTCCTCC |
| H-FASN-QP-F | AAGGACCTGTCTAGGTTTGATGC |
| H-FASN-QP-R | TGGCTTCATAGGTGACTTCCA |
| H-PPARa-QP-F | ATGGTGGACACGGAAAGCC |
| H-PPARa-QP-R | CGATGGATTGCGAAATCTCTTGG |
| H-CIDEC-QP-F | AAGTCCCTTAGCCTTCTCTACC |
| H-CIDEC-QP-R | CCTTCCTCACGCTTCGATCC |
| H-PLIN2-QP-F | ATGGCATCCGTTGCAGTTGAT |
| H-PLIN2-QP-R | GGACATGAGGTCATACGTGGAG |
| H-CD36-QP-F | GGCTGTGACCGGAACTGTG |
| H-CD36-QP-R | AGGTCTCCAACTGGCATTAGAA |
| H-ACOX1-QP-F | ACTCGCAGCCAGCGTTATG |
| H-ACOX1-QP-R | AGGGTCAGCGATGCCAAAC |
| H-CPT1a-QP-F | TCCAGTTGGCTTATCGTGGTG |
| H-CPT1a-QP-R | TCCAGAGTCCGATTGATTTTTGC |
| H-CYP7A1-QP-F | GAGAAGGCAAACGGGTGAAC |
| H-CYP7A1-QP-R | GGATTGGCACCAAATTGCAGA |
| H-ABCG1-QP-F | ATTCAGGGACCTTTCCTATTCGG |
| H-ABCG1-QP-R | CTCACCACTATTGAACTTCCCG |
| H-FATP1-QP-F | GGGGCAGTGTCTCATCTATGG |
| H-FATP1-QP-R | CCGATGTACTGAACCACCGT |
| H-FABP1-QP-F | ATGAGTTTCTCCGGCAAGTACC |
| H-FABP1-QP-R | CTCTTCCGGCAGACCGATTG |
| H-ACC-QP-F | CAAGCCGATCACCAAGAGTAAA |
| H-ACC-QP-R | CCCTGAGTTATCAGAGGCTGG |
| H-PDK4-QP-F | GGAGCATTTCTCGCGCTACA |
| H-PDK4-QP-R | ACAGGCAATTCTTGTCGCAAA |
| H-LCAD-QP-F | AGGGGATCTGTACTCCGCAG |
| H-LCAD-QP-R | CTCTGTCATTGCTATTGCACCA |
| H-MCAD-QP-F | ACAGGGGTTCAGACTGCTATT |
| H-MCAD-QP-R | TCCTCCGTTGGTTATCCACAT |
| H-UCP2-QP-F | CCCCGAAGCCTCTACAATGG |
| H-UCP2-QP-R | CTGAGCTTGGAATCGGACCTT |
| M-SREBP1-QP-F | TGACCCGGCTATTCCGTGA |
| M-SREBP1-QP-R | CTGGGCTGAGCAATACAGTTC |
| M-HMGCR-QP-F | TGTTCACCGGCAACAACAAGA |
| M-HMGCR-QP-R | CCGCGTTATCGTCAGGATGA |
| M-SCD1-QP-F | TTCTTGCGATACACTCTGGTGC |

**Table 2.  Continued**

| Primer name | Primer sequence (5'-3') |
|---|---|
| M-SCD1-QP-R | CGGGATTGAATGTTCTTGTCGT |
| M-FASN-QP-F | AGGTGGTGATAGCCGGTATGT |
| M-FASN-QP-R | TGGGTAATCCATAGAGCCCAG |
| M-PPARa-QP-F | AACATCGAGTGTCGAATATGTGG |
| M-PPARa-QP-R | CCGAATAGTTCGCCGAAAGAA |
| M-CIDEC-QP-F | ATGGACTACGCCATGAAGTCT |
| M-CIDEC-QP-R | CGGTGCTAACACGACAGGG |
| M-PLIN2-QP-F | CTTGTGTCCTCCGCTTATGTC |
| M-PLIN2-QP-R | GCAGAGGTCACGGTCTTCAC |
| M-CD36-QP-F | ATGGGCTGTGATCGGAACTG |
| M-CD36-QP-R | TTTGCCACGTCATCTGGGTTT |
| M-ACOX1-QP-F | CCGCCACCTTCAATCCAGAG |
| M-ACOX1-QP-R | CAAGTTCTCGATTTCTCGACGG |
| M-CPT1a-QP-F | TGGCATCATCACTGGTGTGTT |
| M-CPT1a-QP-R | GTCTAGGGTCCGATTGATCTTTG |
| M-CYP7A1-QP-F | GCTGTGGTAGTGAGCTGTTG |
| M-CYP7A1-QP-R | GTTGTCCAAAGGAGGTTCACC |
| M-ABCG1-QP-F | GTGGATGAGGTTGAGACAGACC |
| M-ABCG1-QP-R | CCTCGGGTACAGAGTAGGAAAG |
| M-FATP1-QP-F | CTGGGACTTCCGTGGACCT |
| M-FATP1-QP-R | TCTTGCAGACGATACGCAGAA |
| M-FABP1-QP-F | ATGAACTTCTCCGGCAAGTACC |
| M-FABP1-QP-R | GGTCCTCGGGCAGACCTAT |
| M-ACC-QP-F | CTCCCGATTCATAATTGGGTCTG |
| M-ACC-QP-R | TCGACCTTGTTTTACTAGGTGC |
| M-LCAD-QP-F | GCTTGGCATCAACATCGCAG |
| M-LCAD-QP-R | ATTCGCAATATAGGGCATGACAA |
| M-MCAD-QP-F | AACACAACACTCGAAAGCGG |
| M-MCAD-QP-R | TTCTGCTGTTCCGTCAACTCA |
| M-UCP2-QP-F | ATGGTTGGTTTCAAGGCCACA |
| M-UCP2-QP-R | TTGGCGGTATCCAGAGGGAA |

## Western blot

Total protein was extracted from tissue and cell samples with RIPA lysis buffer (Beyotime) supplemented with 1 mM phenyl methane sulfonyl fluoride, 4 μl/ml of cocktail, and phosphatase inhibitors. The protein concentration was measured using the BCA Protein Assay Kit (Thermo Fisher Scientific) according to manufacturer's instructions. 20 μg protein was solubilized in sample loading buffer (Beyotime) and heated at 98°C for 12 min. The obtained proteins were separated by using SDS–PAGE electrophoresis and transferred to PVDF membrane (Millipore). Membranes were incubated with indicated primary antibodies at 4°C overnight and HRP-conjugated secondary antibodies at room temperature for 1 h. The chem-iluminescence gel imaging system (Sagecreation) and high-

**Table 3. Antibodies used for Western blot and IHC.**

| Antibody name/lot number | Brand name |
| --- | --- |
| Anti-DKK1 antibody ab93017 | Abcam |
| β-Tubulin antibody #2146 | CST |
| Goat anti-rabbit IgG (H&L) [HRP] KC-RB-035 | Aksomics |
| Anti-CD36 antibody ab133625 | Abcam |
| β-Actin antibody #4970 | CST |
| HRP-GAPDH antibody HRP-60004 | Proteintech |
| Phospho-SAPK/JNK antibody #4668 | CST |
| JNK antibody #9252 | CST |
| Phospho-FoxO1 (Ser256) antibody #9461 | CST |
| FoxO1 (C29H4) rabbit mAb #2880 | CST |
| Phospho-Akt (Ser473) antibody #2443 | CST |
| T-Akt antibody #4691 | CST |
| (p)-PPARγ antibody bs-3737R | Bioss |
| PPARγ antibody #4060 | CST |
| ERK1(pT202/Y204)/ERK2(T185/Y187) MAB1018 | R&D |
| Anti-ERK1/2 MAB1576 | R&D |
| Anti-FAS antibody ab128856 | Abcam |
| Anti-SCD1 antibody ab236868 | Abcam |
| ACC antibody #3676 | CST |
| Anti-SREBP2 antibody ab30682 | Abcam |
| ACOX1 10957-1-AP | Proteintech |
| CPT1α 15184-1-AP | Proteintech |
| Anti-LaminB ab16048 | Abcam |

**Table 4. Primers used for CD36 promoter recombinant plasmids.**

| Primer name | Primer sequence (5′-3′) |
| --- | --- |
| M-CD36-P1138-F | CATGCTAGAAAGTCAAAACCCCTATAACCC |
| M-CD36-Prom-R | GGAAGATCTGCTATTATCTCCTCTCAGTG |

Differences between two groups were evaluated through two-tailed $t$ test. $P$-values were denoted as follows: $*P < 0.05$; $**P < 0.01$; and $***P < 0.001$.

## Data Availability

Gene expression RNA-Seq data have been deposited at the NCBI Gene Expression Omnibus (GEO), accession numbers are GSE197746. All data supporting the findings of this study are available in the article and its supplementary information. The authors declare that additional information is available from the corresponding author upon request.

## Supplementary Information

## Acknowledgements

We thank the First Affiliated Hospital of Jinan University for providing the biopsy sample with liver section. We also thank the outstanding technical support from the animal center and instrument center of Guangzhou Institutes of Biomedicine and Health, Chinese Academy of Sciences. Thanks to Abdul Sammad for helping to polish the article. This study was supported by the National Key R&D Program of China (2019YFA0111300), the Sino-German rapid response funding call for COVID-19 related research (C-0031), National Natural Science Foundation of China (31871379), Guangdong Basic and Applied Basic Research Foundation (2021A1515220095).

### Author Contributions

Z Yang: conceptualization, data curation, software, formal analysis, validation, investigation, and writing—original draft, review, and editing.
X Huang: data curation, validation, and methodology.
J Zhang: software and writing—original draft.
K You: validation, investigation, and methodology.
Y Xiong: resources, software, and methodology.
J Fang: resources and data curation.
A Getachew: software and formal analysis.
Z Cheng: investigation and methodology.
X Yu: investigation and methodology.
Y Wang: investigation and methodology.
F Wu: formal analysis, investigation, and methodology.
N Wang: investigation and methodology.
S Feng: resources and data curation.
X Lin: software and funding acquisition.

sensitivity ECL Kit (Thermo Fisher Scientific) were used to detect the protein expression level. Cellular cytoplasmic and nucleic proteins were obtained by using nucleic and cytoplasmic protein extraction kit (Beyotime) according to the manufacturer's instructions. The antibodies used in this study are listed in Table 3.

### Luciferase reporter assay

The *CD36* promoter (−1,138 to +165 bp) was PCR-amplified from mouse genomic DNA by using primers listed in Table 4. The validated sequences were cloned into a PGL4-basic vector by using the NheI and BglII restriction sites. Empty pGL4 plasmid was used as a negative control plasmid (NC-promoter). For the luciferase reporter assays, HepG2 cells were transfected with pGL4-CD36-promoter or pGL4-NC-promoter using Lipofectamine 3000 (Invitrogen) following the manufacturer's instructions. After co-transfection of the GFP or DKK1 OE plasmid for 48 h, the luciferase activities were measured with a Dual Luciferase Reporter Assay system (Promega) according to the manufacturer's instructions.

### Data analyses

All data were analyzed using GraphPad Prism 7 (GraphPad Software, Inc.). The quantitative data are presented as the means ± SD.

F Yang: software and funding acquisition.

Y Chen: data curation, investigation, and methodology.

H Wei: resources and funding acquisition.

Y-X Li: supervision, funding acquisition, validation, visualization, project administration, and writing—review and editing.

## Conflict of Interest Statement

The authors declare that they have no conflicts of interest.

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
