## [Reviewer comments · Life Science Alliance]

Life Science Alliance

Hepatic DKK1 driven steatosis is CD36-dependent

Zhen Yang, Xinping Huang, Jiaye Zhang, Kai You, Yue Xiong, Ji Fang, Anteneh Getachew, Ziqi Cheng, Xiaorui Yu, Yan Wang, Feima Wu, Ning Wang, Shufen Feng, Xianhua Lin, Fan Yang, Yan Chen, Hongcheng Wei, and Yinxiong Li

DOI: <https://doi.org/10.26508/lsa.202201665>

Corresponding author(s): *Yinxiong Li, Guangzhou Institutes of Biomedicine and Health*

Review Timeline:

Submission Date:	2022-08-11
Editorial Decision:	2022-09-16
Revision Received:	2022-10-08
Editorial Decision:	2022-10-25
Revision Received:	2022-10-27
Accepted:	2022-10-27

Scientific Editor: Novella Guidi

Transaction Report:

September 16, 2022

Re: Life Science Alliance manuscript #LSA-2022-01665-T

Prof. Yinxiong Li
Guangzhou Institutes of Biomedicine and Health
South China Institute for Stem Cell Biology a
190 kaiyuan avenue
Guangzhou 510530
China

Dear Dr. Li,

Thank you for submitting your manuscript entitled "Hepatic DKK1 driven steatosis is CD36-dependent" to Life Science Alliance. The manuscript was assessed by expert reviewers, whose comments are appended to this letter. We invite you to submit a revised manuscript addressing the Reviewer comments.

Thank you for this interesting contribution to Life Science Alliance. We are looking forward to receiving your revised manuscript.

Sincerely,

B. MANUSCRIPT ORGANIZATION AND FORMATTING:

Reviewer #1 (Comments to the Authors (Required)):

Yang et al. reported the role of DKK1 in Non-alcoholic fatty liver disease (NAFLD). They have shown that DKK1 works in CD36 dependent manner and is involved in insulin resistance through the ERK-PPAR γ -CD36 pathway. The authors observed that DKK1 had been upregulated in NAFLD liver tissues. This observation was validated in mice model, AML-12 and HepG2 cell lines by over and loss in expression using an adenovirus delivery system. The in vitro and in vivo models were treated with fatty acids, and the expression of DKK1 was analysed. Further, the DKK1 CD36 axis was studied using CD36 knockout AML-12 and HepG2 cell lines. These cell lines were treated with a fatty acid, and no significant changes in fatty acid accumulation were observed.

The findings and the mechanistic study were well designed and defined. However, minor corrections are required.

1. In histological images, secondary controls are missing.
2. In Fig:1 (F) and Fig:4 (C), (D), (E), (F), Relative mRNA expression has been mentioned on Y-axis. However, the Fig:4 (C), (D), (E), and (F), the controls have been considered as 1, which is Fold change calculation using the delta ct method, but in Fig:1 (F), the controls have not been considered as 1. Please explain if there are any differences in the calculation, as the term "Relative mRNA expression" has not been apparent
3. In Fig:1 (F), Please replot the graph to show individual values. E.g., as demonstrated in Fig:1 (G)

Reviewer #2 (Comments to the Authors (Required)):

In this manuscript, Zhen Yang and colleagues describe their studies showing that DKK1 promotes NAFLD through CD36 dependent pathways. The authors found that DKK1 increases the capability of hepatocytes to uptake fatty acids and this activity of DKK1 operates through ERK-PPARG-CD36 axis. These data are obtained using a variety of biological systems including patients with NAFLD, animal models of fatty liver and cultured hepatoblastoma cells. This work is innovative and the results are convincing. However, there are several issues that need to be addressed.

Comments:

- 1) The authors need to present better quality data and accurate statements regarding several observations. Figure 1 A and B show results of staining of DKK1 in patients with NAFLD and in HFD-treated mice. The authors state that DKK1 "was primary localized in the steatosis hepatocytes" (Fig 1A). First, as it is seen on the figure, the intensive staining was observed in a fibrotic area, but it is not clear if staining is inside of hepatocytes or outside. A better quality images with higher magnification should be presented to support this statement. It would be also necessary to include examples from several NAFLD patients. Second, similar improvements should be done for Figure 1B which shows results of DKK1 staining in HFD-treated mice.
- 2) Data for differences in liver steatosis in HFD treated mice (Figure 2) need an additional support. Results of the H&E and Oil Red-O staining need to be backed up by examination of markers of steatosis using QRT-PCR approach.
- 3) The quality of some data in Figure 5 is poor. First, the presented Western blot images for SREBP2 and SCD1 (Fig 5A) need to be improved. Second, differences for CD36 protein should be presented as ratios to b-tubulin control. Third, better images should be presented for p-PPAR γ and PPAR γ blots on Figure 5H.
- 4) Minor. The manuscript needs editing.

Thank you and the reviewers for your time and the constructive suggestions and comments, followed those suggestions, we performed more experiments and revised our manuscript accordingly.

Reviewer #1 (Comments to the Authors (Required)):

Yang et al. reported the role of DKK1 in Non-alcoholic fatty liver disease (NAFLD). They have shown that DKK1 works in CD36 dependent manner and is involved in insulin resistance through the ERK-PPAR γ -CD36 pathway. The authors observed that DKK1 had been upregulated in NAFLD liver tissues. This observation was validated in mice model, AML-12 and HepG2 cell lines by over and loss in expression using an adenovirus delivery system. The in vitro and in vivo models were treated with fatty acids, and the expression of DKK1 was analysed. Further, the DKK1 CD36 axis was studied using CD36 knockout AML-12 and HepG2 cell lines. These cell lines were treated with a fatty acid, and no significant changes in fatty acid accumulation were observed.

The findings and the mechanistic study were well designed and defined. However, minor corrections are required.

1. In histological images, secondary controls are missing.

Response: Many thanks for your encouragement and the suggestion for adding the represented images of secondary controls. Indeed, we carefully designed this experiment with a whole set of control groups, and the results of histological images were summarized in the following figure (Figure S3A).

There were eight groups in total to be conducted for the DKK1 expression manipulation experiments in which there were parallel four groups for chow or high fat diet condition, including control (first control), AAV-GFP-NC (secondary control), AAV-OE-DKK1 and AAV-sh-DKK1, correspond to none virus infection, virus expressing GFP, virus overexpressing DKK1 and virus expressing ShRNA for knockdown DKK1, respectively.

In chow condition, there was none steatosis or no significant difference among those four groups, while in HFD condition, all four groups were induced steatosis with ballooning hepatocytes at different degrees. Both HFD (first control) and HFD-AAV-GFP-NC (secondary control) revealed certain level steatosis without significant difference, however, overexpressed DKK1 promoted the steatosis, furthermore, knockdown the expression of DKK1 significantly alleviated the steatosis under HFD condition, and the result was confirmed by TG measurement in liver. In the revised version, we add this figure S3A in the supplementary materials.

The AAV infection did not cause steatosis in chow condition, and the steatosis had no significant difference with or without AAV-GFP infection under HFD condition. Based on these two facts, considering the size and content in one figure, therefore, we decide to pick up four panels of represented images and organize the Figure 2A, including those

groups of chow-AAV-GFP-NC, HFD-AAV-GFP-NC, HFD-AAV-OE-DKK1 and HFD-sh-DKK1 group mice to present our results.

In addition, we replaced the images in Figure 2D with better histological quality and higher magnification images.

2. In Fig:1 (F) and Fig:4 (C), (D), (E), (F), Relative mRNA expression has been mentioned on Y-axis. However, the Fig:4 (C), (D), (E), and (F), the controls have been considered as 1, which is Fold change calculation using the delta ct method, but in Fig:1 (F), the controls have not been considered as 1. Please explain if there are any differences in the calculation, as the term "Relative mRNA expression" has not been apparent

Response: Followed this suggestion, in the revised version of this manuscript, we unified the "relative mRNA expression" on Y-axis of all qPCR data as the fold changes comparing to the controls.

3. In Fig:1 (F), Please replot the graph to show individual values. E.g., as demonstrated in Fig:1 (G)

Response: Yes, we re-plotted the data of Figure 1F with individual values.

Reviewer #2 (Comments to the Authors (Required):

In this manuscript, Zhen Yang and colleagues describe their studies showing that DKK1 promotes NAFLD through CD36 dependent pathways. The authors found that DKK1 increases the capability of hepatocytes to uptake fatty acids and this activity of DKK1 operates through ERK-PPARG-CD36 axis. These data are obtained using a variety of biological systems including patients with NAFLD, animal models of fatty liver and cultured hepatoblastoma cells. This work is innovative and the results are convincing. However, there are several issues that need to be addressed.

Comments:

1) The authors need to present better quality data and accurate statements regarding several observations. Figure 1 A and B show results of staining of DKK1 in patients with NAFLD and in HFD-treated mice. The authors state that DKK1 "was primary localized in the steatosis hepatocytes" (Fig 1A). First, as it is seen on the figure, the intensive staining

was observed in a fibrotic area, but it is not clear if staining is inside of hepatocytes or outside. A better quality images with higher magnification should be presented to support this statement. It would be also necessary to include examples from several NAFLD patients. Second, similar improvements should be done for Figure 1B which shows results of DKK1 staining in HFD-treated mice.

Response: Thank you for this constructive suggestion. We have revised the Figure 1A and B accordingly.

And we added more samples from NAFLD patients IHC results were shown in the figure S1A.

Although DKK1 staining results demonstrated that DKK1 expression was significantly elevated in NAFLD livers, it is true that “the intensive staining was observed in a fibrotic area, but it is not clear if staining is inside of hepatocytes or outside”, since the intensive steatosis and cell death. Therefore, we conducted to isolate and separate the different cell types directly from the liver of HFD mice, then performed Western blot analysis, the result clearly revealed that even DKK1 was expressed in some low degree in HSC and LSEC, but the majority resource of DKK1 was from the expression of hepatocytes under HFD condition (Figure 1G).

2) Data for differences in liver steatosis in HFD treated mice (Figure 2) need an additional

support. Results of the H&E and Oil Red-O staining need to be backed up by examination of markers of steatosis using QRT-PCR approach.

Response: Yes, followed this suggestion, we performed more analysis on those HFD-induced liver samples. The total triglyceride and total cholesterol contents of liver were measured (Figure 2E, F), and further analyzed 8 markers of steatosis (lipid metabolism related genes) using QRT-PCR approach to support the steatosis levels in liver under different DKK1 gene manipulations (Figure 2I).

3) The quality of some data in Figure 5 is poor. First, the presented Western blot images for SREBP2 and SCD1 (Fig 5A) need to be improved. Second, differences for CD36 protein should be presented as ratios to b-tubulin control. Third, better images should be presented for p-PPAR γ and PPAR γ blots on Figure 5H.

Response: To assure the accuracy of the result, followed the three suggestions from this reviewer, we conducted to repeat those experiments, and the data was updated in Figure 5.

4) Minor. The manuscript needs editing.

Response: As suggested, we carefully went through the whole manuscript for grammatical and spelling errors, then we invited a colleague who is a native English speaker to edit and proofread the revised version of our manuscript.

October 25, 2022

RE: Life Science Alliance Manuscript #LSA-2022-01665-TR

Prof. Yinxiong Li
Guangzhou Institutes of Biomedicine and Health
South China Institute for Stem Cell Biology a
190 kaiyuan avenue
Guangzhou 510530
China

Dear Dr. Li,

Thank you for submitting your revised manuscript entitled "Hepatic DKK1 driven steatosis is CD36-dependent". We would be happy to publish your paper in Life Science Alliance pending final revisions necessary to meet our formatting guidelines.

- please address the remaining Reviewer 1 point
- please add a separate figure legend section to your main manuscript
- please upload your table files as editable doc or excel files or make sure that they are included in the doc file of your main manuscript text

Figure Check:

- the scale bars in Figures 1,2,3,5, S1, S2 and S3 should be more visible. We suggest to use a white font with bolder lines.

A. FINAL FILES:

B. MANUSCRIPT ORGANIZATION AND FORMATTING:

Sincerely,

Reviewer #1 (Comments to the Authors (Required)):

The authors incorporated all the suggestions which are satisfactory; in Fig 1(E) the fold change term is used in Y-axis, however, the calculations reflected as same as previous data which is not clear. If the analysis was done using the Fold change, the control group (Chow diet) should be assigned as 1, currently, it is shown as 0.6. Please correct it by removing the term fold change as this is western blot bands intensity calculation, not mRNA qPCR analysis or provide an explanation for the calculation used to plot the graph.

Reviewer #2 (Comments to the Authors (Required)):

The authors have adequately addressed my comments as well as comments of other reviewers. The revised manuscript will be of great interest for field of NAFLD.

The final revised version was edited with the guide from you and the suggestion from reviewer 1. We sincerely express our thanks to all of you for your help and time, the interaction among the editor, reviewer and the authors makes this work to be a proud piece in our academic career.

Figure Check:

-the scale bars in Figures 1,2,3,5, S1, S2 and S3 should be more visible. We suggest to use a white font with bolder lines.

Yes, your suggestions were fully executed in this submitted revised version.

Reviewer #1 (Comments to the Authors (Required)):

The authors incorporated all the suggestions which are satisfactory; in Fig 1(E) the fold change term is used in Y-axis, however, the calculations reflected as same as previous data which is not clear. If the analysis was done using the Fold change, the control group (Chow diet) should be assigned as 1, currently, it is shown as 0.6. Please correct it by removing the term fold change as this is western blot bands intensity calculation, not mRNA qPCR analysis or provide an explanation for the calculation used to plot the graph.

Response: Many thanks for your constructive guide and suggestions. Followed your suggestion, in Figures 1E, we changed the "fold change" to "Relative DKK1 levels" as the precise meaning.

Reviewer #2 (Comments to the Authors (Required)):

The authors have adequately addressed my comments as well as comments of other

reviewers. The revised manuscript will be of great interest for field of NAFLD.

Response: Many thanks for your time and help, we are more than happy to work with you, and this revised version contains so many contributions from your kindness and guidance that make it reaches to a higher academic level.

October 27, 2022

RE: Life Science Alliance Manuscript #LSA-2022-01665-TRR

Prof. Yinxiong Li
Guangzhou Institutes of Biomedicine and Health
South China Institute for Stem Cell Biology a
190 kaiyuan avenue
Guangzhou 510530
China

Dear Dr. Li,

Thank you for submitting your Research Article entitled "Hepatic DKK1 driven steatosis is CD36-dependent". It is a pleasure to let you know that your manuscript is now accepted for publication in Life Science Alliance. Congratulations on this interesting work.

DISTRIBUTION OF MATERIALS:

Again, congratulations on a very nice paper. I hope you found the review process to be constructive and are pleased with how the manuscript was handled editorially. We look forward to future exciting submissions from your lab.

Sincerely,
